# Control Mechanisms of the Tumor Suppressor PDCD4: Expression and Functions

**DOI:** 10.3390/ijms20092304

**Published:** 2019-05-09

**Authors:** Sachiko Matsuhashi, M. Manirujjaman, Hiroshi Hamajima, Iwata Ozaki

**Affiliations:** 1Department of Internal Medicine, Saga Medical School, Saga University, 5-1-1 Nabeshima, Saga 849-8501, Japan; monirbio31@gmail.com; 2Saga Food & Cosmetics Laboratory, Division of Food Manufacturing Industry Promotion, SAGA Regional Industry Support Center, 114 Yaemizo, Nabesima-Machi, Saga 849-0932, Japan; hamajimahiroshi@gmail.com; 3Health Administration Center, Saga Medical School, Saga University, 5-1-1 Nabeshima, Saga 849-8501, Japan; ozaki@cc.saga-u.ac.jp

**Keywords:** tumor suppressor, *PDCD4*, tumor promotor, EGF, TPA, carcinogenesis

## Abstract

*PDCD4* is a novel tumor suppressor to show multi-functions inhibiting cell growth, tumor invasion, metastasis, and inducing apoptosis. PDCD4 protein binds to the translation initiation factor eIF4A, some transcription factors, and many other factors and modulates the function of the binding partners. PDCD4 downregulation stimulates and PDCD4 upregulation inhibits the TPA-induced transformation of cells. However, *PDCD4* gene mutations have not been found in tumor cells but gene expression was post transcriptionally downregulated by micro environmental factors such as growth factors and interleukins. In this review, we focus on the suppression mechanisms of PDCD4 protein that is induced by the tumor promotors EGF and TPA, and in the inflammatory conditions. PDCD4-protein is phosphorylated at 2 serines in the SCF^βTRCP^ ubiquitin ligase binding sequences via EGF and/or TPA induced signaling pathway, ubiquitinated, by the ubiquitin ligase and degraded in the proteasome system. The PDCD4 protein synthesis is inhibited by microRNAs including miR21.

## 1. Introduction

12-O-tetradecanoylphorbol-13-acetate (TPA) and epidermal growth factor (EGF) are well known potent promotors for carcinogenesis [1,2]. Colburn et al. established the JB6 mouse epidermal cell lines, which are promotion sensitive (P+) and resistant (P-) to TPA-induced neoplastic transformation [3]. Activation protein-1 (AP-1), a trans-acting transcription factor, was shown to be stimulated in the promotion sensitive (P+), but not in the promotion resistant (P-) cells [4,5]. Experiments to find a suppressor of the neoplastic transformation induced by the tumor promotors resulted in Pdcd4 suppressing AP-1 activation and TPA-induced neoplastic transformation in the JB6 cells [6]. Since then, studies on *PDCD4* have been accelerated and it becomes clear that *PDCD4* is a multifunctional tumor suppressor [7] (Figure 1).

Human *PDCD4* was first found as a nuclear antigen gene and mapped at chromosome 10q24 [7]. The expression of the gene was modulated by interleukins such as IL-2, IL-12, and IL-15 [8]. The mouse homolog was isolated as a gene that expression was up-regulated in the apoptosis [9] or down-regulated by topoisomerase inhibitors such as Camptothecin [10]. Rat [11], chicken [12], frog [13], fishes [14], and Drosophila [15] homologs were also isolated. The gene homolog is present in plants but not in yeasts [16]. The gene is well conserved in species; for example, the homology between human and fish (elephant shark) is 72 % identical in the protein level.

*PDCD4* expresses ubiquitously in the tissues and the protein is localized in the nuclei or cytoplasm or in both depending on the cell types. The expression of PDCD4-protein is often down-regulated in the tumors [17,18,19,20] but is maintained in high levels in a large number of tumors.

## 2. The Function of *PDCD4*

### 2.1. PDCD4 Inhibits Neoplastic Transformation

Cmarik et al. [6] reported that *Pdcd4* expression is down-regulated in mouse JB6 promotion sensitive (P+) cells compared to resistant (P-) cells. The JB6 P+ and JB6 P- cells are genetic variants that differ in their transformation response to tumor promotors. In response to the tumor promotor TPA, P+ cells form colonies in agar plates, but P- cells do not. They showed that the down-regulation of Pdcd4 in P- cells resulted in acquisition of transformation sensitive phenotype (P+) and the up-regulation of Pdcd4 in the P+ type cells changed to P- type cells. *Pdcd4* knock-down mice normally develop. However, the *Pdcd4* deficient mice develop spontaneous lymphomas, mostly the B lymphoma origin and have a significantly reduced life span compared with wild type siblings [21]. Jansen et al. [22] generated transgenic mice overexpressing *Pdcd4* in the epidermis that the Pdcd4 expression is regulated under the K14-promotor. The skin of K14-regulated mice shows significant reduction in papilloma formation, carcinoma incidence, and papilloma to carcinoma conversion frequency in response to the tumor promotor TPA, compared with wild-type mice. These results indicate that Pdcd4 functions as the inhibitor of neoplastic transformation [22].

### 2.2. PDCD4 Controls Translation

PDCD4-protein contains two MA3 domains homologous to the M1 domain of eukaryotic translation initiation factor 4G (eIF4G) [23] (Figure 2). The eIF4G is a scaffold protein and interacts with the cap binding protein eukaryotic translation initiation factor 4E (eIF4E) and eukaryotic translation initiation factor 4A (eIF4A). The eIF4A binds to the M1 domain of eIF4G [23,24]. Thereby, the eIF4A is included as a member in the translation initiation factor complex eukaryotic translation initiation factor 4F (eIF4F), functions as the RNA helicase to linearize the secondary structures in the 5′-untranslated region (5′-UTR) of mRNA and stimulated the cap-dependent translation. PDCD4 protein binds eIF4A through the MA3 domains, interferes the eIF4A binding to eIF4G in the translation initiation complex and inhibits the RNA helicase activity of eIF4A and cap-dependent translations [24] (Figure 3). It was reported that the mRNAs of the tumor suppressor TP53 and stress-activated-protein kinase interacting protein-1 (SIN1) are such physiological translational targets of PDCD4 [25,26].

The role of PDCD4-protein in the regulation mechanisms of translation is more complex. PDCD4-protein contains one RNA binding site at the N-terminal region [27] (Figure 2) and inhibits some protein elongation binding to special secondary RNA structures in the coding region of the mRNA (Figure 3). It was shown that PDCD4-protein interacts with the coding region of *c-Myb* and *A-Myb* mRNA through the RNA binding domain and inhibits the translation of the mRNA [28,29]. Besides the cap-dependent translation initiation site, protein synthesis is initiated from the special secondary structure located in the 5′-UTR internal ribosomal entry site (IRES). Usually, PDCD4 does not inhibit the IRES-dependent translation. However, it is shown that PDCD4 binds to the IRES of the anti-apoptotic proteins X chromosome linked inhibitor of apoptosis (XIAP) and B-cell lymphoma-extra-large (Bcl-XL) mRNAs and inhibits the translation of the mRNAs [30]. Further, PDCD4-protein is a binding partner of the poly (A) binding protein (PABP) that binds to the poly A tail of mRNA and control protein synthesis interacting with the translation initiation complex [31]. The PABP may interact with the PDCD4-protein bound to IRES or the secondary structures in the coding region of mRNA and inhibit the translation of the mRNA.

### 2.3. PDCD4 Knockdown Activates AP-1 and β-catenin/Tcf-dependent Transcriptions

Besides the inhibition of protein synthesis, PDCD4 suppresses AP-1 transactivation that activates gene expression to stimulate cell proliferation, transformation, tumor invasion, and metastasis [24]. The transcription factor AP-1 is a homodimer or heterodimer of JUN and FOS [32] and is activated by the up-stream kinases Jun N-terminal kinase (JNK) to activate JUN and mitogen-activated protein kinase kinase kinase kinase 1 (MAP4K1) to activate JNK [33]. PDCD4 was shown to inhibit the activation of c-Jun by suppressing the JNK-activation in avian QT6 cells [34] and to suppress the expression of the upstream kinase *MAP4K1* in RKO human colon carcinoma cells [33]. It was shown that a c-Myc binding site is located in the *MAP4K1* gene promoter region and MAP4K1 expression is stimulated by c-Myc [35]. On the other hand, *PDCD4* knockdown inhibits E-cadherin expression up-regulating the zinc-finger transcription factor *SNAIL* [36]. The SNAIL is a transcription repressor and binds to the E-boxes (CAGGTG) of E-cadherin promotor and inhibits the transcription of E-cadherin [37]. The down-regulation of E-cadherin activates β-catenin/T-cell specific transcription factor (Tcf)-dependent transcription, stimulating the expression of *C-MYC*, one of the target gens [36]. Altogether, *PDCD4* knockdown upregulates *C-MYC* expression via SNAIL-E-cadherin-β-catenin/Tcf-*C-MYC* pathway and in turn the C-MYC may stimulate *MAP4K1* expression and activate the downstream factors JNK and AP-1.

### 2.4. Inhibition of Transcriptions

It has been reported that PDCD4-protein associates with many kinds of transcription factors and controls transcriptions. For examples, PDCD4 interacts directly with the transcription factor TWIST 1 and leads to the inhibition of cell growth via the down-regulation of the TWIST 1 target gene Y-box binding protein-1 (*YB-1*) [38]. PDCD4 binds to the DNA-binding domain of TWIST 1, inhibits the DNA binding ability of the protein and down-regulates the *YB-1* expression. The gene of urokinase receptor (*u-PAR*) which promotes tumor invasion/metastasis contains Sp1 and Sp3 transcription factor binding sites in the promotor region. It was shown that PDCD4 interacts with Sp1 and Sp3 inhibiting the *u-PAR* expression and tumor invasion/intravasion [39]. Baker et al. reported that PDCD4 inhibits nuclear factor-kappaB (NF-κB)-dependent transcription by directly interacting with the p65 subunit of the factor in human glioma cells [40]. The key mediator of Notch signaling CSL (CBF1, SUH, Lag 1) functions as the repressor of transcriptions under basal conditions and is activated by Notch signaling. PDCD4 was shown to be a member in the complex of CSL repressor and to function as the repressor of the target genes [41].

### 2.5. Interaction with Cytoplasmic Factors

PDCD4-protein also interacts with cytoplasmic factors. One of such factors is the PABP [31] as mentioned in the section of protein synthesis control by PDCD4. Besides PABP, death domain associated protein (DAXX), a scaffold protein with roles in diverse process is shown to be an interactive partner of PDCD4 protein. DAXX serves as a scaffold for the serine/threonine protein kinase homeodomain-interacting protein kinase 2 (HIPK2) and the tumor suppressor p53 (TP53) and stimulates the phosphorylation of TP53 at Ser 46 in response to DNA damages. PDCD4 inhibits the phosphorylation of TP53 in the absence of DNA damage disrupting the DAXX and HIPK2 association, while DNA damages decrease the PDCD4 expression and increase the phosphorylation of TP53 protein. Thus, PDCD4 may function to restrain TP53 activity in unstressed conditions [42].

### 2.6. Induction of Apoptosis

Regarding apoptosis induction, *PDCD4* expression is upregulated and PDCD4-protein is accumulated in the cell nuclei as cells undergo apoptosis [18]. PDCD4 over expression activates the pro-apoptotic member of Bcl2 protein family *BAX* followed by the release of cytochrome C from mitochondria and activation of caspases 8, 9, and 3, resulting in the induction of apoptosis in hepatoma cells, although the mechanism of *BAX* activation has not been elucidated [18]. It has been reported that *PDCD4* knockdown also induces apoptosis. Eto et al. demonstrated that the loss of PDCD4 up-regulated pro-caspase 3 expression resulting in the elevation of active caspase 3 levels and the induction of apoptosis without apoptotic stimuli in HeLa cells [43]. We have observed that siRNA-mediated *PDCD4* knockdown suppressed cell growth and induced senescence inhibiting retinoblastoma (RB)-protein (pRB)-phosphorylation via down-regulating the expression of pRB itself and cyclin dependent kinases (CDKs) which phosphorylate pRB and up-regulating the CDK inhibitor p21(CDKNI4) [44]. *RB* is a central regulator of cell cycle progression. pRB binds to E2F transcription factor and arrests cell cycle progression. On the phosphorylation of pRB by CDK4/6 and CDK2, E2F is released and activates the gene expression required for cell proliferation [45]. *CDKNI4* knockdown rescued the senescence and cell death as well as the inhibition of pRb-phosphorylation and CDK expression induced by *PDCD4* knockdown in the hepatoma cells [44]. *CDKNI4* expression is generally induced by TP53 transcription factor in response to DNA damages. However, *PDCD4* knockdown also stimulated *CDKNI4* expression in *TP53*-deficient cells and siRNA-mediated *TP53* knockdown cells, indicating that *PDCD4* knockdown induces *CDKNI4* expression in TP53-independent manner. The results indicate that PDCD4 is an important cell cycle regulator and may contribute to the control of cell cycle via p21(CDKNI4) in hepatoma cells.

Alessio et al. reported that silencing of *RB* gene induced senescence of human mesenchymal stromal cells (MSCs), but not of human fibroblasts, mouse MSCs, and mouse fibroblasts. Interestingly, the senescence is significantly reduced in mouse fibroblasts by the *RB*-silencing. The results indicate that RB might play different roles in senescence induction depending on cell types and species [46].

Guo et al demonstrated that the modulation of CDKs expression and cell death patterns induced by *PDCD4* knockdown, are different between the wild type hepatoma HepG2 and the mutant Hep3B cells, which is *RB*- and *TP53*-defficient [44]. Kang et al. reported that PDCD4 is upregulated in senescent human fibroblasts [47]. Conversely, *PDCD4* knockdown stimulated cell proliferation by upregulating cyclin D1 expression in HT29 colon tumor cells [48]. The results indicate that the role of PDCD4 in the RB-pathway might be different depending on the context and cell types.

Finally, the functions of PDCD4 are summarized in Table 1. Although the functions are diverse, it can be seen that most functions are associated with anti-tumor activities.

## 3. Regulation of *PDCD4* Expression

*PDCD4* expression is controlled at transcription, translation, and protein degradation levels.

### 3.1. Controls at the Transcription

The expression of *PDCD4* was shown to be controlled by zinc-finger protein (ZBP) transcription factors such as specificity protein (Sp) and ZBP-89 [49]. There are several potential binding sites for the ZBPs, especially for Sp family and ZBP-89 in the *PDCD4* promotor region. Leupold et al. [49] identified four Sp1/Sp3/Sp4-binding elements that are indispensable for basal promotor activity but the overexpression of Sp1 and Sp3 was not sufficient to enhance *PDCD4* expression and ZBP-89 is also needed for *PDCD4* expression. ZBP-89 enhances *PDCD4* expression by binding to the basal promotor region either alone or by interacting with Sp family members [49]. Schlichter et al. identified myeloblastosis (Myb) transcription factor binding sites in the promotor region of chicken *Pdcd4* gene and demonstrated that *Pdcd4* expression is induced by Myb using a luciferase reporter plasmid with the promotor region of chicken *Pdcd4* [12,50].

TGF-β1 treatment of Huh7 hepatoma cells increased PDCD4 expression up-regulating *PDCD4* mRNA expression through the TGF-β1-activated Smad signaling pathway and induced apoptosis [18]. PDCD4-protein was accumulated in the nuclei on the TGF-β1-treatment of the cells and apoptotic cells were increased in parallel with the increase of PDCD4-protein in the nuclei [18].

*PDCD4* mRNA levels are decreased in many cancer tissues and cell lines as compared with the corresponding normal tissues [17,19,20]. The transcription of *PDCD4* gene is shown to be down-regulated by the methylation of 5′CpG islands in the *PDCD4* promoter region. Gao et al. observed that the methylation of PDCD4 5′CpG islands was significantly correlated with the loss of *PDCD4* mRNA expression in 47% (14/30) of glioma tissues [19]. They demonstrated that blocking the methylation in glioma cells using the DNA methyltransferase inhibitor 5-aza-2 deoxycytidine restored *PDCD4* expression and inhibited the proliferation of the cells [19].

### 3.2. The Inhibition of PDCD4 Translation by miRNAs

MicroRNAs (miRNAs), endogenous non-coding small RNAs regulate the stability or translational efficiency by binding to specific sites on the 3′-UTR of target messenger RNAs. They are important regulators in processes such as differentiation, proliferation, and inhibiting apoptosis [51]. It is predicted more than 1000 miRNAs are coded on the human genome. Primary transcripts of miRNA (pri-miRNA) are cleaved by the RNase III endonuclease Dorsha and its dsRNA binding partner Pasha, leading to pre-miRNAs and processed further to 20–24 nt-miRNAs via Dicer, another RNase III [51]. Among the miRNAs, miR21 is widely expressed in the tissues, up-regulated in many kinds of cancers and plays key roles in human cancers associating with cell cycle, cancer progression, advanced tumor stages, and poor survival rate [51,52,53]. The mir21 displays oncogenic activity and is classified as an oncomir [54,55].

The translation of *PDCD4* mRNA is inhibited by many kinds of miRNAs including miR21. The miR21 target site is located at nt233-349 of *PDCD4* 3′-UTR region. It was demonstrated that using a Luciferase reporter plasmid containing the predicted miR21 target sequence of *PDCD4* mRNA at the 3′-region of Luciferase gene, the reporter activity was decreased in the presence of miR21 but unaffected as the reporter construct was mutated at the miR21 target site [56]. The activity was also decreased when miR21 was inhibited by LNA-derived oligonucleotide inhibitor [57]. These results indicate that the miR21 targets *PDCD4* mRNA and post-transcriptionally down-regulates *PDCD4* expression, resulting in the increased transformation, invasion, and metastasis of cancer cells [56], Besides miR21, many other kinds of microRNA such as miR182, miR16, miR150, miR499 and other microRNAs has been shown to target the *PDCD4* mRNA down-regulating PDCD4 expression in different cellular systems [58,59,60,61]. Zhang et al. [61] reported that both miR21 and miR499 are highly expressed in tonsil squamous cell carcinoma (SCC) tissues down-regulating PDCD4 expression. The 3ʹUTR of *PDCD4* mRNA contains one mir21 target site and three miR499 target sites. They constructed two *PDCD4* plasmids with or without the 3ʹUTR region, investigated *PDCD4* expression of the plasmids in the presence or absence of miR21 and/or miR499 in HEK-293 cells and demonstrated that the initial suppression of PDCD4 expression was mediated by miR21, while the sustained suppression was mediated by miR499 [61].

### 3.3. Controls at Protein Degradation

On the serum treatment of cells, PDCD4-protein levels are down-regulated. Dorrello et al. demonstrated that the PDCD4-protein is phosphorylated by ribosomal protein S6 kinase 1 (S6K1) downstream of the serum mitogen-induced phosphatidylinositol 3-kinase (PI3K)-Akt (protein kinase B)-mechanistic Target of Rapamycin (mTOR) signaling pathway (Figure 4). The phosphorylated protein is then ubiquitinated and degraded in proteasomes [62]. Schmid et al. reported that the tumor promotor TPA decreased PDCD4 protein levels in mouse skin keratinocytes as well as in human HEK293 cells [63]. TPA activated PKC-dependent PI3K-Akt-mTOR-S6K pathway and mitogen-activated protein/extracellular signal-regulated kinase (ERK) kinase (MEK)-ERK signaling pathway in the HEK293 cells [63]. PDCD4-protein was then phosphorylated by both Akt and p70S6K, ubiquitinated and degraded in proteasomes. They show that MEK-ERK signaling facilitated the substantial degradation of the protein in proteasomes in the cells [63].

### 3.4. Arginine Methylation of PDCD4 Protein

PDCD4-protein levels are usually down-regulated in cancer cells, but substantial numbers of patients with high PDCD4-protein levels in tumors show poor survival. It was shown that the methylation of PDCD4 at 110R by arginine methyltransferase 5 (PRMT5) increased the PDCD4 levels, tumor cell growth and viability in a nutritional deprivation condition [64]. The arginine methylation may inhibit PDCD4-protein functions and allow the tumor to grow more aggressively.

## 4. The Tumor Promotors EGF and TPA Induce PDCD4 Degradation in Huh7 Hepatoma Cells

It is known that transforming growth factor-β1 (TGF-β1) induces apoptosis in the hepatocytes and some hepatoma cell lines [65]. Epidermal growth factor (EGF) is a potent growth promoting factor for hepatocytes and inhibits the TGF-β1-induced apoptosis [2]. It was demonstrated that the expression of PDCD4 increases in cultured Huh7 hepatoma cells exposed to TGF-β1 and that ectopic PDCD4 overexpression induces such cells to undergo apoptosis [18]. The results indicate that TGF-β1 may induce apoptosis via PDCD4 overexpression in the Huh 7 hepatoma cells. EGF suppresses the TGF-β1-induced BAX activation, mitochondria events, caspase cascade activation, and PDCD4 expression [66]. The results indicate that EGF suppresses TGF-β1-induced apoptosis by decreasing the PDCD4 expression. The EGF-induced PDCD4 decrease is inhibited by the inhibitors of PI3K and mTOR and by the proteasome inhibitor MG132 [66] (Figure 4). *S6K1* knockdown was shown to increase PDCD4-protein levels [67]. These results suggest that PDCD4-protein may be phosphorylated through the EGF-activated PI3K-Akt-mTOR-S6k1 signaling pathway and degraded in the proteasome system (Figure 4).

TPA also inhibits the TGF-β1-induced apoptosis down-regulating PDCD4 expression but does not stimulate the phosphorylation of Akt and S6K1. The mTOR inhibitor rapamycin does not inhibit the TPA-induced PDCD4 suppression in Huh7 hepatoma cells [68] (Figure 5). The PKC (PRCK)*α*, PRCKδ and PRCKε of the PRCKs that are activated by TPA, are expressed in the Huh7 cells. siRNA-mediated knockdown of the *PRCK*s showed that *PRCKδ* and *PRCKε* knockdown up-regulated, but *PRCKα* did not change PDCD4 expression [68]. From these results, it was concluded that EGF and TPA suppress PDCD4-protein levels through different signaling pathways in Huh7 cells.

Experiments to identify new substrates of the Skp1-Cul1-Fbox protein (SCF)^βTRCP^ ubiquitin ligase revealed that PDCD4-protein is a substrate of the enzyme [62]. PDCD4-protein contains a canonical β-transducin repeat containing protein (βTRCP)-binding motif, D^70^SGRGDS^76^, corresponding to the consensus sequence DSGXX(X)S (Figure 6A). The phosphorylation of both serines in the degron sequence is needed to allow recognition by βTRCP [69]. It is known that phosphorylation on residues surrounding the degron sequence promotes the phosphorylation of the two serines in the degron in many βTRCP substrates [70]. In the case of PDCD4-protein, the phosphorylation at Ser67 promotes the phosphorylation at Ser71 and Ser76 in the degron [62].

A sequence R62LRKNS67 corresponding to the canonical S6K1 phosphorylation motif consensus sequence K/RXRXXS [70], is localized at N-terminal side of the degron sequence in the human PDCD4-protein (Figure 6A). In response to mitogens, S6K1 is activated and phosphorylates Ser67 on PDCD4. This event, in turn, promotes the phosphorylation of Ser71 and Ser76 in the degron and allows binding PDCD4-protein to βTRCP [62]. PDCD4-protein is then ubiquitinated by the SCF^βTRCP^ ubiquitin ligase and degraded in the proteasomes [62] (Figure 4).

## 5. Ser71 and Ser76 Are Phosphorylated by Different Enzymes

Ser67 (S67) is phosphorylated by S6K1, which is activated with mTOR, and then the phosphorylation at S67 triggers further phosphorylation at Ser71 (S71) and Ser76 (S76) followed by the degradation of PDCD4-protein in the ubiquitin proteasome system as mentioned above [62]. To determine whether the phosphorylation of S67, S71, and S76 are involved in the EGF- or TPA-induced degradation of PDCD4-protein, we constructed green fluorescent protein (GFP)-tagged *PDCD4* plasmid with thymidine kinase (TK)-promotor because the cytomegalovirus (CMV)-promoter is responsive to EGF and TPA stimulation [66]. Using the TK-promoter plasmid, *PDCD4* mutant plasmids were created such that serine was replaced with alanine (A) or glutamic acid (D) at S67 (S67/A or S67/D), S71 (S71/A or S71/D) and S76 (S76/A or S76/D) (Figure 5). Duplicated mutants (S67/71/A, S67/76/A and S71/76/A or S67/71/D, S67/76/D and S71/76/D) and a triplicated mutant S67/71/76/A were also prepared. The plasmids were transfected into Huh7 cells. The transfected cells were cultured in the presence or absence of EGF or TPA with or without the inhibitor rapamycin or pan-PKC inhibitor. The mutant *PDCD4* expression in the cells was assayed by Western blotting and the results are summarized in Figure 5. EGF showed little effect on the expression of S67/A mutant *PDCD4* as expected, while TPA down-regulated the expression of the mutant [66]. Both mitogens showed no effect on the expression of both S71/A and S76/A mutants. From the results, it is concluded that the phosphorylation at S67 is necessary for the EGF-induced degradation of PDCD4-protein but is not necessary for the TPA-induced degradation of the protein. Namely, TPA may be able to induce the phosphorylation of both S71 and S76 independently from the phosphorylation of S67.

On the other hand, the expression levels of the PDCD4 mutants replaced with glutamic acid (D) S67/D, S71/D and S76/D were low in the absence of inhibitors but upregulated by the treatment with MG132 proteasome inhibitor similarly to S67/A mutants. The results indicated that the all mutants can be degraded in the ubiquitin-proteasome system and therefore the replacement of serine to glutamic acid mimics the phosphorylation of serine at S67, S71 and S76. The mTOR inhibitor rapamycin increased both the protein level of S67/D and S71/D mutants while the reagent did not increase the protein level of S76/D mutant. EGF suppressed the expression of S71/D mutant, but not that of S67/D and S76/D mutants, in the presence of pan-PKC inhibitor [66]. The results showed that S76 is phosphorylated by EGF-induced signaling pathway and the phosphorylation of S76 is inhibited by rapamycin indicating that S76 is phosphorylated by S6K1, a substrate of mTOR.

TPA suppresses the expression of all the S/D mutants both in the presence and absence of rapamycin (Figure 5, unpublished data) indicating that TPA does not activate the PKC-dependent PI3K-Akt-mTOR-S6K1 signaling pathway in the Huh7 cells. Altogether, these results indicated that all these phospho-mimic mutants are degraded in the ubiquitin-proteasome system and that Ser76 may be phosphorylated by S6K1 which is activated through EGF-induced PI3K-Akt-mTOR signaling pathway, as well as S67 and S457, and that S71 and S76 may be phosphorylated by a member of PKC signaling pathway different from S6K1 (Figure 7).

It was shown that the consensus sequence of p90 ribosomal S6 kinase (RSK) is RXRXXS/T as same as that of p70S6K1 and the enzyme phosphorylates, the Ser76 and Ser457, as well as Ser67 of human PDCD4-protein in melanoma cells [71]. The phosphorylation of PDCD4 at S76 by RSK is also reported by Cuesta and Holz using the Triple Negative Breast Cancer (TNBC) MDA-MB-231 cells [72]. RSK is a MAPK-activated protein kinase required for melanoma proliferation. Galan et al. [71] used a quantitative phosphorproteomics approach to identify phosphorylation substrates of RSK, defined the RSK consensus phosphorylation motif and found that the sequence is significantly overlapped with the binding consensus sequence of 14-3-3. They showed that 14-3-3 protein was co-immuno-precipitated with PDCD4-protein when RSK was activated by the PMA treatment of melanoma cells and conclude that 14-3-3 protein interacted with PDCD4 protein phosphorylated at S67, S76, and/or S457 sequestering the PDCD4-protein into the cytoplasm and stimulated the degradation of the protein in the proteasome system [71].

RSK is activated via mitogen-induced mitogen activated protein kinase (MAPK) pathway [71]. We have observed that the pathway is activated by EGF and TPA but the MAPK inhibitor PD98059 does not inhibit the EGF- or TPA-induced suppression of PDCD4-protein levels in Huh7 cells [66]. Therefore, we concluded that RSK expression is low and the involvement of RSK in the PDCD4 protein degradation is not significant in the Huh7 cells.

## 6. EGF Down-Regulates *PDCD4* mRNA Levels but TPA Does Not

*PDCD4* mRNA levels were also down-regulated by EGF treatments. We have observed rapamycin inhibited the EGF-induced down-regulation of *PDCD4* mRNA levels in Huh7 hepatoma cells [66]. Vihreva et al. reported that the transcription of *PDCD4* was repressed in mTOR-signaling dependent manner, in lung cancer cells [73]. However, the mRNA levels were not changed by TPA treatments in the hepatoma cells [66]. The results indicate that EGF contributes to downregulate *PDCD4* expression at both levels of transcription and protein degradation and that *PDCD4* transcription is controlled by mTOR or its downstream signaling pathway.

Altogether, it is concluded that S67 and S76 of the three serines involved in the degradation mechanism of PDCD4-protein, are phosphorylated by S6K1 activated via EGF-induced PI3K-Akt-mTOR-S6K1 signaling pathway and S71 and S76 may be phosphorylated independently from the phosphorylation of S67 by TPA mediated signaling pathway in Huh7 hepatoma cells (Figure 7). The PDCD4-protein phosphorylated at S71 and S76 is then ubiquitinated by SCF^βTRCP^ ubiquitin ligase and degraded in proteasomes resulting in the stimulation of cell proliferation and tumor progression. The sequence upstream S76 R69DSGRGDS76 (RXXXRXXS) is different from the canonical consensus sequence RXRXXS of S6K1 substrate, but there is homology between the sequences around the three phosphorylation sites S67, S76 and S457 by S6K1 in PDCD4 protein, as shown in Figure 6B. Thus, the induction mechanism of PDCD4 degradation explains, at least partly, the mechanism of EGF or TPA function as a tumor promotor.

## 7. Inflammation and Carcinogenesis

Recently, many investigators have reported that PDCD4 is involved in the induction of inflammation. *Pdcd4* knockdown mice developed spontaneous tumors, mostly B-lymphoma and significantly shorter life spans than wild type siblings, but they are resistant to the induction of inflammatory diseases such as autoimmune encephalomyelitis and diabetes [21]. *PDCD4* may be able to regulate both carcinogenesis and inflammation.

The macrophages derived from human U937 monocytes by TPA treatment increased the expression and secretion of inflammatory cytokines such as tumor necrosis factor-α (TNF-α), interleukin 6 (IL6) and interleukin 8 (IL8) [74]. Lipopolysaccharide (LPS) also stimulates RAW264.7 mouse macrophages to produce a similar environmental condition [75]. The conditional medium of the activated macrophages downregulated PDCD4-protein levels, activating the PI3K-Akt-mTOR-S6K signaling pathway and phosphorylating PDCD4-protein in various tumor cells [74,75].

Sheedy et al. reported that the LPS-treatment of RAW264.7 mouse macrophages up-regulated *Pdcd4* expression, but it was abolished at 24 h of the treatment accompanying with high levels of miR21 expression [76]. The induction of miR21 expression required the Toll-like receptor (TLR) adaptor protein Myo88 and nuclear factor-kappa B (NF-κB). The NF-kB activity increased in a *Pdcd4* dependent manner at the early response to LPS. Further, TLR ligands such as Pam3CSk4 (TLR2 ligand) or poly(L:C) (TLR3 ligand) up-regulated *Pdcd4* expression [76]. Similar results were also reported by using primary bone marrow-derived macrophages (BMDMs) [76]. These results indicated that TLR signaling up-regulated *Pdcd4* expression, induced miR21 expression via Pdcd4-NF-κB signaling and then miR21 inhibits the translation of *Pdcd4* mRNA, resulting in the loss of Pdcd4 tumor suppressor (Figure 8). PDCD4 suppression then induces the expression of IL-10 and TNFα [21,74,75]. NF-κB is a transcription factor consisting of p65 (REL)-p50 dimers and inactivated by binding with inhibitor-κB (I-κB). When I-κB kinase (IκK) is activated by pro-inflammatory cytokines such as TNFα, the activated IκK phosphorylates I-κB and activates NF-κB, which is then translocated into nuclei [77]. The promotor region of *MIR21* gene contains NF-κB binding sites [78] as well as other miRNA genes such as *MIR16* [78] or *MIR34a* [79] and the activated NF-κB stimulates the expression of *MIR21* binding to the sites. NF-κB is also reported to activate the expression of cytokines such as IL6 and IL8 [80] (Figure 8).

Cigarette smoking is a risk factor for cancers and nicotine was reported to promote tumor growth. Ng et al. reported that miR16 and miR21 were significantly up-regulated in nicotine treated gastric cancer cells and that nicotine mediated I-κB degradation and the nuclear translocation of NF-κB via prostaglandinF2 (PGF-2) signaling resulting in the up-regulation of the miR16 and miR21 [78].

Members of IL6 family are involved in a variety of biological responses such as immune response, hematopoiesis, cell growth, differentiation, as well as inflammation. IL6 activates the Janus kinase (JAK)/Signal transducer and activator of transcription-3 (STAT3) pathway through binding to the IL6 receptor [81] (Figure 8). The STAT3 transcription factor also regulates microRNA gene expression. Rozouski et al. identified putative STAT3 binding sites in the promotor region of 160 miRNA genes including *MIR21* [82]. It is reported that the STAT3 directly activates *MIR21* and *MIR181b-1,* respectively, suppressing the targets phosphatase and tensin homolog (PTEN) and CYLD (CYLD lysine 63 deubiquitinase) tumor suppressors [83,84].

The matrix proteoglycan decorin (*DCN*) is an early response gene evoked by septic inflammation and increases in septic patients or in mice with LPS-induced sepsis. Merline et al. reported that decorin controls inflammation and tumor growth through PDCD4-miR21 axis [85]. Recently, it was shown that PDCD4 is also an important factor for inflammatory diseases such as asthma [86] and atherosclerosis [87,88].

PDCD4 is a novel tumor suppressor that mutations have not been reported in cancer cells, but the protein levels are post transcriptionally controlled by environmental factors such as growth factors, interleukins, and TPA. Thus, in the inflammatory areas, inflammatory cytokines downregulated PDCD4-protein levels in surrounding cells via NF-κB-miR21 and/or STAT3-miR21 axis (Figure 8) and by the degradation of the protein resulting in high-risk environmental conditions for tumor promotion and carcinogenesis.

## 8. Clinical Aspects

The anti-tumor activity of PDCD4 is well documented in invitro systems especially by using colon cancer cell lines [24,39] and in Pdcd4 transgenic and knockdown mice [21,22]. Ectopic PDCD4 expression induces apoptosis of tumor cells so far as we have tested.

*PDCD4* expression levels are downregulated in many kinds of tumor tissues obtained from patients, as shown in Table 2. Further, the loss of PDCD4 expression is correlated with poor prognosis and survival in breast cancer [89], colorectal cancer [90], lung cancer [17], ovarian cancer [91] and renal cell carcinoma [92] (Table 2). The data indicate that *PDCD4* may be a promising target for the cancer prevention and therapy. The increase of PDCD4-protein levels may be effective for the purposes.

Several compounds are found to up-regulate PDCD4-protein levels (Table 3). Curcumin and resveratrol are natural polyphenol compounds, which show ant-oxidant, anti-tumor, or anti-inflammatory activities and foods containing the compounds are good for the health. Both compounds up-regulate PDCD4-protein levels by inhibiting mir21 expression [93,94,95,96]. WP1066, a STAT3 inhibitor also up-regulate the protein levels by inhibiting the STAT3/miR21 axis [97].

Schmid group scientists have tried to find compounds to stabilize PDCD4-protein by inhibiting the PDCD4-protein degradation in the ubiquitin-proteasome system. They have identified diaryl disulfides [98] and isolated tricyclic guanidine alkaloids from the marine animal sponge [99] as the PDCD4-protein stabilizer (Table 3).

MicroRNAs to target *PDCD4* mRNA also may be a useful tool for the upregulation of PDCD4. Among them, nucleotide anti-mir21 drugs inhibit colon cancer cell metastasis up-regulating PDCD4-protein levels in in vitro experiments [100]. The anti-mir21 drug has already cleared phase 1 clinical trial in Polycystic kidney disease including Alport syndrome patients [101]. *MIR429* might also be useful for the therapy as it upregulates PDCD4-protein level. [102]

*PDCD4* cDNA and *PDCD4* specific siRNA also might be useful drugs for the cancer therapy. Huang et al. delivered *Pdcd4* cDNA into cancerous mouse lung by aerosol delivery and observed the inhibition of tumor cell proliferation and induction of tumor cell apoptosis [103]. *PDCD4* specific siRNA also induces tumor cell death [43,44].

It has been reported that PDCD4 upregulation sensitizes tumor cells to anti-tumor drugs such as cisplatin [104], gemcitabine [102], geldanamycin and tamoxifen [105]. The combination therapy of these anti-tumor drugs and the compounds to upregulate PDCD4-protein levels, may be useful tools for cancer treatments.

The targets of mir21 are not only *PDCD4* but also other tumor suppressors such as *PTEN* mRNAs. However, PDCD4 should be at least partly involved in the anti-tumor activities of anti-mir21 drug or other drugs to upregulate PDCD4 via inhibition of miR21 expression. The clinical trial targeting *PDCD4*, including pre-clinical trials, is largely in the future problems.

## 9. Future Perspectives and Open Questions

PDCD4 is an important factor in the inflammation besides in cancer prevention. In addition to the clinical trial in cancer prevention and therapy, the investigation on the PDCD4 function in the inflammatory diseases such as asthma, atherosclerosis or diabetes may also be the future interests.

PDCD4 gene is well conserved in species but its function is diverse and might be different in context and cell types. It is an interesting problem whether the functions are the same or different among species. It should be elucidated that PDCD4 functions as a translation inhibitor or tumor suppressor in all livings containing the gene.

## Figures and Tables

**Figure 1 ijms-20-02304-f001:**
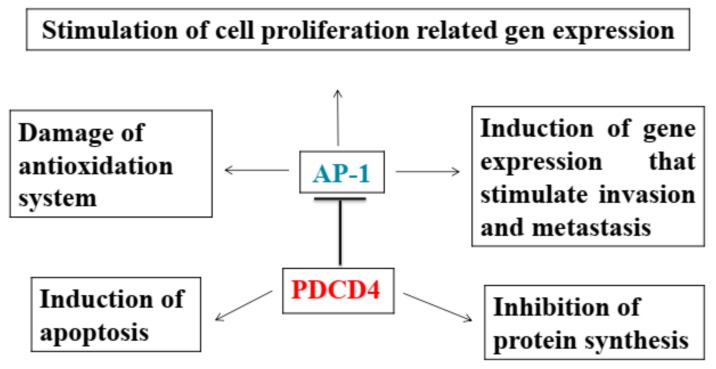
PDCD4, a tumor suppressor, suppresses carcinogenesis by multifunction.

**Figure 2 ijms-20-02304-f002:**
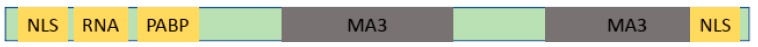
Schematic structure of PDCD4-protein. NLS, Nuclear localization signal; RNA, RNA binding site; PABP, poly(A) binding protein (PABP) binding site; MA3, MA3 domain homologous to eIF4G M1 domain. PDCD4 protein interacts with eIF4A through the MA3 domain and inhibits cap-dependent protein synthesis.

**Figure 3 ijms-20-02304-f003:**
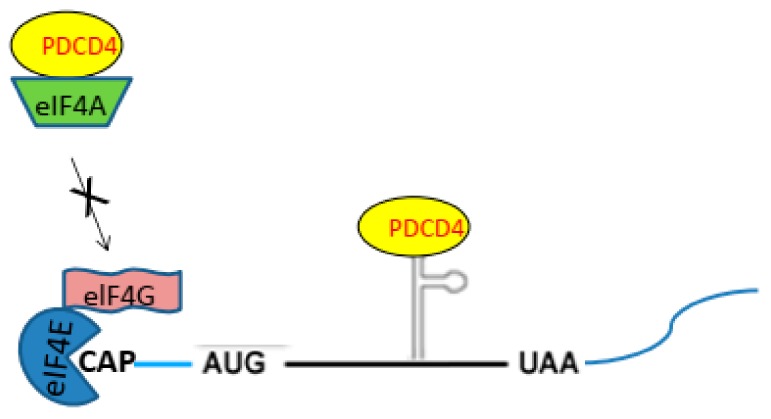
Controls of translation by PDCD4. PDCD4 binds to eIF4A and inhibits its helicase activity suppressing the protein synthesis. Also, PDCD4 binds to the special secondary structure of mRNA and inhibits the protein synthesis.

**Figure 4 ijms-20-02304-f004:**
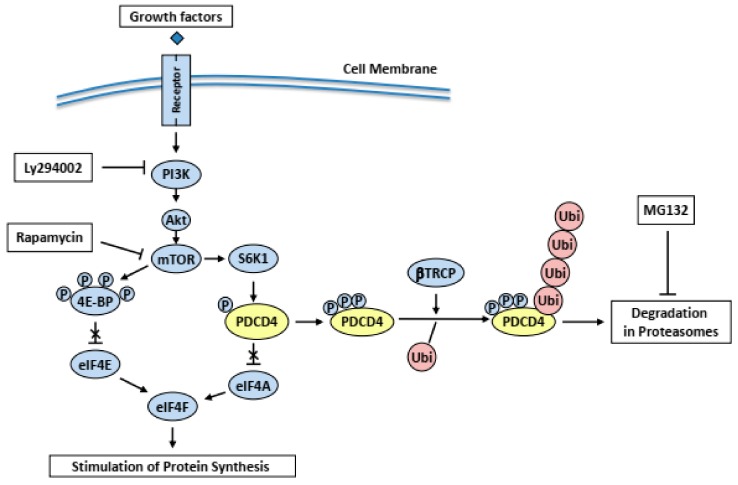
Pathways of EGF mediated protein synthesis and PDCD4 degradation. EGF stimulates the phosphorylation of the eIF4E inhibitor 4E-BP via activation of PI3K-Akt-mTOR pathway and thereby activating protein synthesis. On the other hand, mTOR-activated S6K1 phosphorylates PDCD4-protein. The phosphorylated protein is ubiquitinated by SCF^βTRCP^ (βTRCP) ubiquitin (Ubi) ligase and degraded in proteasomes stimulating protein synthesis.

**Figure 5 ijms-20-02304-f005:**
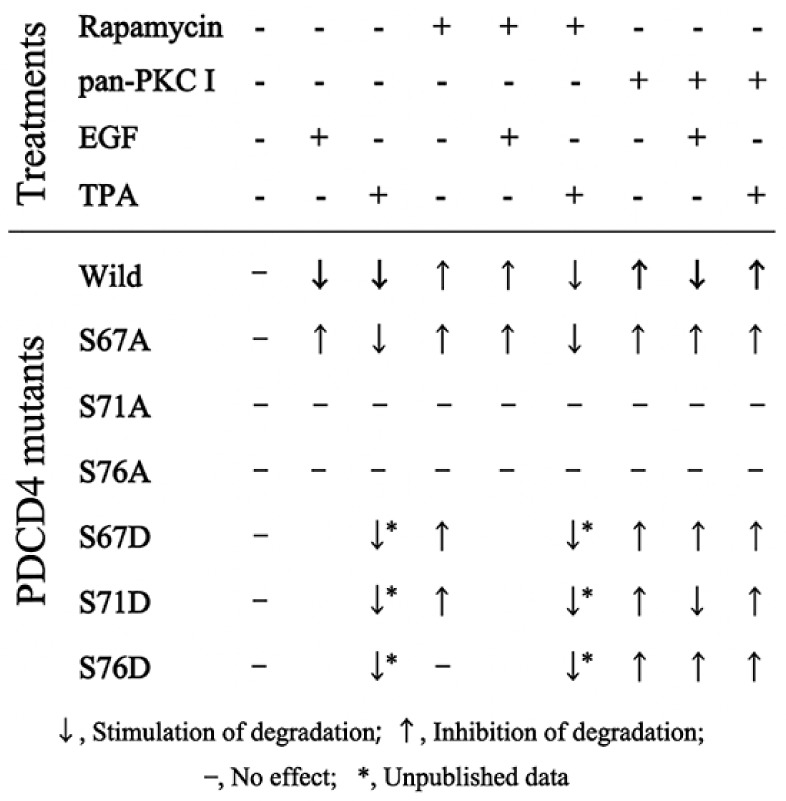
Expression patterns of PDCD4 mutants in the presence or absence of inhibitors with or without tumor promoters. Huh7 cells transfected with the mutant PDCD4-GFP plasmids were treated with or without EGF or TPA in the presence or absence of rapamycin or pan-PKC inhibitor (pan-PKCI). The expression of ectopic mutant PDCD4 conjugated with GFP was analyzed by Western blot.

**Figure 6 ijms-20-02304-f006:**
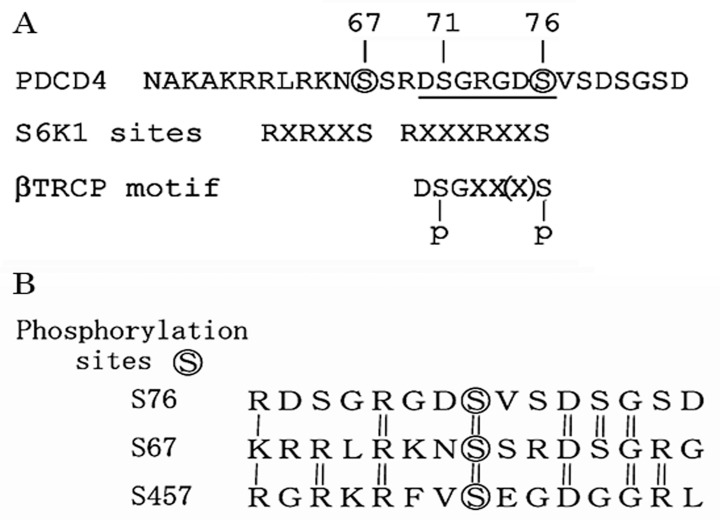
S6K1 phosphorylation sites and β-TRCP binding motif in the PDCD4 protein. (**A**) Serines (S) phosphorylated by S6K1 are marked with circles and SCF^βTRCP^ binding sequence is underlined in the PDCD4 protein sequence. RXRXXS, Canonical S6K1 substrate consensus sequence. (**B**) Homology of the amino acid sequence around the three S6K1 phosphorylation sites in the PDCD4 protein.

**Figure 7 ijms-20-02304-f007:**
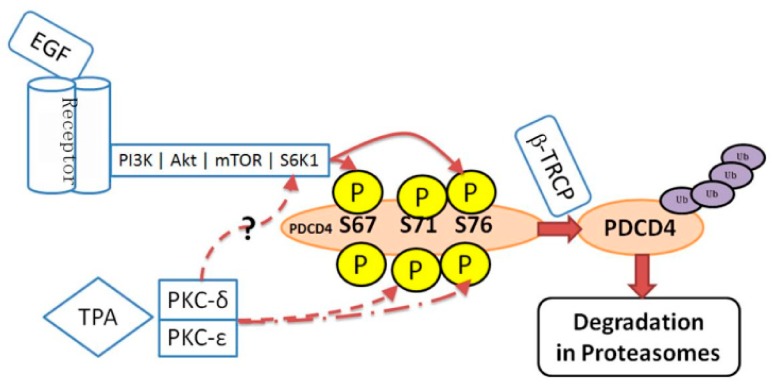
EGF and TPA mediated pathway to phosphorylate serines in the SCF^βTRCP^ recognition site of PDCD4-protein in Huh7 hepatoma cells. EGF activates PI3K-Akt-mTOR-S6K1 signaling pathway and the activated S6K1 phosphorylates PDCD4-protein at S67 and S76. TPA-mediated pathway via PKC-δ and/or PKC-ε may phosphorylate S71 and S76. PKC-dependent activation of S6K1 was not observed in Huh7 hepatoma cells.

**Figure 8 ijms-20-02304-f008:**
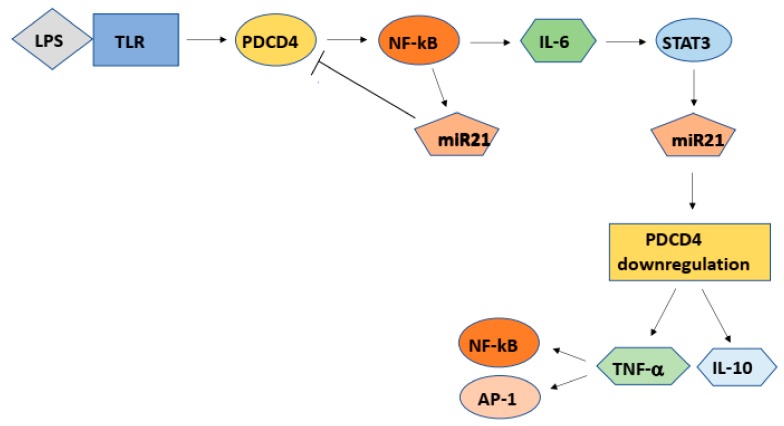
PDCD4 expression is suppressed by miR21 via NF-κB and/or STAT3 in the cells in the inflammatory area.

**Table 1 ijms-20-02304-t001:** Molecular Functions of PDCD4.

Function of PDCD4	Experimental System or Target	Effects of PDCD4	Ref.
Neoplastic transformation inhibitor	JB6 mouse epidermal cells	Cells with high Pdcd4 levels resist to TPA induced neoplastic transformation	[6]
	*Pdcd4* knockout mice	Induction of spontaneous B-lymphoma	[21]
	transgenic mice with epidermal specific *Pdcd4* expression	The epidermis resists to TPA-induced carcinogenesis	[22]
Inhibition of translation	SIN1 cap-dependent translation	Inhibition of colon carcinoma invasion	[26]
	TP53 cap-dependent translation	Maintenance of TP53 levels	[25]
	XIAP and Bcl-X_1_ IRES-mediated translation	Stimulation of apoptosis	[30]
	CMyb and AMyb protein elongation inhibition	A regulatory feedback loop, Myb is a transcription factor to stimulate *PDCD4* expression	[28,29]
Inhibition of AP-1 activation	Via inhibition of the upstream kinase *MAP4K1* suppression	Suppression of human colon carcinoma cell invasion	[33]
Maintenance of E-cadherin levels	*PDCD4* knockdown down-regulated E-cadherin expression via *SNAIL*.	Suppression of invasion via reduction of catenin/Tcf-dependent transcription.	[36]
Inhibition of transcription factors	PDCD4 binds to TWIST DNA binding site	Suppression of cell growth downregulating the target gene *YB1* expression	[38]
	Sp family; PDCD4 binds to Sp1/Sp3 and inhibits *u-PAR* transcription	Suppression of colon carcinoma cell invasion/intravasion	[39]
	NF-κB; PDCD4 binds with p65 subunit	Inhibition of glioma cell growth	[40]
	PDCD4 is a part of CSL transcription inhibitor complex	Negative control of stromal fibroblast conversion into cancer associated fibroblasts	[41]
Binding with cytoplasmic protein	DAXX, a scaffold protein with roles in diverse processes	Controls of the activity of DAXX binding partner proteins	[42]
Induction of apoptosis	Overexpression of PDCD4	Apoptosis induction of tumor cells	[18]
	*PDCD4* knockdown	Induction of senescence and/or apoptosis	[43,44]

**Table 2 ijms-20-02304-t002:** Anti-tumor activity of PDCD4.

Tumors	Roles of PDCD4	Ref.
Brest cancer	PDCD4 overexpression sensitizes aromatase inhibitor (AI)-resistant cells to AI and PDCD4 downregulation is associated with a lower survival of patient in estrogen receptor positive breast cancer.	[89]
Colon carcinoma	PDCD4 overexpression inhibits cancer cell invasion/intravation.	[24,39]
Colorectal cancer	PDCD4 expression is downregulated in the tumor tissues and the loss of PDCD4 correlated with patient survival.	[90]
Epidermal tumor	Epidermis of transgenic mice with epidermis specific *Pdcd4* expression resists to TPA-induced carcinogenesis.	[22]
Glioma cells	*PDCD4* expression is suppressed by the methylation of CpG islands in the promoter region of tumor tissues. The loss of PDCD4 is associated with poor prognosis.	[19]
Hepatoma cells	PDCD4 expression is suppressed in the hepatoma tissues from patients, and PDCD4 overexpression induces apoptosis of hepatoma cells.	[18]
Lung cancer	PDCD4 expression is downregulated in the tumor tissues and the loss of PDCD4 correlated with poor prognosis in the patients.	[17]
B-lymphoma	*Pdcd4* knockout mice induce spontaneous B-lymphoma.	[21]
Ovarian cancer	PDCD4 expression in the tumor cells suppresses the malignant phenotype. The loss of PDCD4 is correlated with patient poor survival.	[91]
Renal cell carcinoma (RCC)	PDCD4 expression is downregulated in RCC tumor tissues and correlated to RCC stage, grade, metastasis and survival.	[92]

**Table 3 ijms-20-02304-t003:** Possible anti-tumor drugs associated with PDCD4.

Drug	Function and Pre-Clinical and Clinical Trials	Ref.
Curcumin	Upregulation of PDCD4-protein levels by inhibiting mir21 expression	[93,94]
Resveratrol	Upregulation of PDCD4-protein levels via Akt/mir21 inhibition. Resveratrol has used for clinical trials	[95,96]
WP1066 (STAT3 inhibitor)	Upregulation of PDCD4-protein levels via inhibition of STAT3-mir21 axis	[97]
Diaryl disulfides	Stabilization of PDCD4-protein by inhibiting the degradation in ubiquitin-proteasome system	[98]
Tricyclic guanidine alkaloids from the marine sponge *Acanthella cavernosa*	Stabilization of PDCD4-protein by inhibiting the degradation in ubiquitin-proteasome system	[99]
Anti-mir21 nucleotide	Stimulation of PDCD4-protein synthesisAn anti-mir21 drug has used for clinical trial in polycystic kidney disease	[100,101]
Mir429	Ectopic expression increased cellular sensitivity to gemcitabine via PDCD4 upregulation	[102]
*PDCD4* cDNA	Ectopic expression of *PDCD4* induces tumor cell apoptosisAerosol delivery of *Pdcd4* cDNA into a mouse lung with tumors inhibits cell proliferation and induces apoptosis of the tumor cells	[18,103]
*PDCD4* specific siRNA	siRNA-mediated *PDCD4* knockdown induces tumor cell death	[43,44]

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
