# Peer review of "Control Mechanisms of the Tumor Suppressor PDCD4: Expression and Functions"

_ijms, 2019, doi:10.3390/ijms20092304_

Round 1
Reviewer 1 Report
The authors presented an interesting review on the Suppression Mechanisms of the Tumor Suppressor PDCD4.
The authors well described studies showing role of PDCD4 in senescence. They cited a paper of Guo et al. showing that PDCD4 Knockdown Induces Senescence in Hepatoma Cells by Up-Regulating the p21 Expression. Moreover, they described a research they performed showing that siRNA-mediated PDCD4 knockdown suppressed cell growth and induced senescence inhibiting retinoblastoma (Rb)-protein (pRb)-phosphorylation via down-regulating the expression of Rb itself and cyclin dependent kinases (CDKs) which phosphorylate pRb and up-regulating the CDK inhibitor p21.
Anyway, role of PDCD4 in senescence may be context dependent. For example, Kang et al. showed the Up-regulation of PDCD4 in senescent human diploid fibroblasts (Biochem Biophys Res Commun. 2002 Apr 26;293(1):617-21. DOI: 10.1016/S0006-291X(02)00264-4). Indeed, also the RB pathway that is associated with PDCD4 may have different roles in senescence, depending on cell type, animal species and experimental conditions (Neoplasia. 2017 Oct;19(10):781-790. doi: 10.1016/j.neo.2017.06.005).
I suggest to address these issues to have a complete vision of PDCD4 role in cell biology.
Author Response
Manuscript ID: 486443
Comments of reviewer 1
Anyway, role of PDCD4 in senescence may be context dependent. For example, Kang et al. showed the Up-regulation of PDCD4 in senescent human diploid fibroblasts (Biochem Biophys Res Commun. 2002 Apr 26;293(1):617-21. DOI: 10.1016/S0006-291X(02)00264-4). Indeed, also the RB pathway that is associated with PDCD4 may have different roles in senescence, depending on cell type, animal species and experimental conditions (Neoplasia. 2017 Oct;19(10):781-790. doi: 10.1016/j.neo.2017.06.005).
I suggest to address these issues to have a complete vision of PDCD4 role in cell biology
Reply
Thank you for your valuable comments. We are interested very much in the Neoplasia paper.
We have added discussion at the end of the subsection 2.6. Induction of apoptosis using the papers by Alessio et al. and Kang et al. according to your comments, in the revised manuscript. The part of addition is highlighted by red color
Reviewer 2 Report
In the current manuscript entitled “Suppression Mechanisms of the Tumor Suppressor PDCD4 Mediated by the Tumor Promoters EGF and TPA” by Sachiko Matsuhashi and colleagues have reviewed the existing literature on the tumor suppressive role of PDCD4 on TPA induced cancer cells transformation. Authors have reviewed the literature on the inhibition of neoplastic transformation, translation control, regulation of transcription, interaction with other proteins, induction of apoptosis by PDCD4. Further, authors have also presented the information related to the regulation of PDCD4 by transcription, translation, protein degradation, phosphorylation and miRNA mediated changes. Further, environmental factors in particular inflammation mediated influence of PDCD4 function is described. This is an exciting review article, but few changes are required before getting into a conclusion. Tittle should be reframed as it is creating some confusion. Separate table related to the following: (A) describing the function of PDCD4, phenotype outcome and the corresponding references etc; (B) drugs are available to activate the function of PDCD4? (C) clinical trials there are involved in PDCD4 activation? (D) How many cancer types PDCD4 acts as a tumor suppressor? (E) regulation of PDCD4 by miRNAs? And a separate para describing the feature directions of PDCD4 research?
Author Response
Manuscript ID: 486443
Comments of reviewer 2
Tittle should be reframed as it is creating some confusion. Separate table related to the following: (A) describing the function of PDCD4, phenotype outcome and the corresponding references etc; (B) drugs are available to activate the function of PDCD4? (C) clinical trials there are involved in PDCD4 activation? (D) How many cancer types PDCD4 acts as a tumor suppressor? (E) regulation of PDCD4 by miRNAs? And a separate para describing the feature directions of PDCD4 research?
Thank you for your valuable comments. Followings are our reply to the comments.
Comment; Tittle should be reframed as it is creating some confusion.
Reply; Tittle is changed to Control mechanisms of the tumor suppressor PDCD4: Expression and functions.
Comment; Separate table related to the following: (A) describing the function of PDCD4, phenotype outcome and the corresponding references etc;
Reply; We have prepared Table 1. Molecular functions of PDCD4 and added it with short discussion at the end of the Section 2. The function of PDCD4.
Comments; (B) drugs are available to activate the function of PDCD4? (C) clinical trials there are involved in PDCD4 activation? (D) How many cancer types PDCD4 acts as a tumor suppressor? (E) regulation of PDCD4 by miRNAs?
Reply; We have prepared two Tables, Table 2. Anti-tumor activity of PDCD4 and Table 3. Possible anti-tumor drugs associated with PDCD4 and described in the new section 8. Clinical aspects.
Comment; And a separate para describing the feature directions of PDCD4 research?
Reply; We have described them in the new section 9. Future problems
In the revised manuscript the part of addition is highlighted by red color
Round 2
Reviewer 1 Report
None.
Author Response
Thank you very much.